Review Article

# Epimutations: raw material for evolution?

Nabeel S Ganem & Peter Sarkies ✉

## Abstract

Epigenetics is fundamental to cell differentiation as it enables cells with identical genomes to adopt distinct fates. Some epigenetic information can also be transmitted between generations, in a process known as transgenerational epigenetic inheritance. This means that potentially epigenetic differences between individuals could contribute to diversity and thus be acted upon by evolution. These epigenetic differences are termed epimutations by analogy to the well-characterized DNA sequence mutations that underpin the standard model of evolution. Here, we evaluate the properties of epimutation, discussing their rate, genome-wide distribution, stability, and effects. Focusing on epimutations in animals, particularly the nematode *C. elegans*, we explore how epimutations compare to DNA sequence mutations in their potential to influence the processes of drift and natural selection that characterize evolution.

**Keywords** Epigenetics; Evolution; Small Non-coding RNAs; Chromatin; Population Genetics
**Subject Categories** Chromatin, Transcription & Genomics; Evolution & Ecology; RNA Biology

## Introduction

While classical genetics holds that traits are passed through generations solely through DNA, the past decade has revealed that non-genetic information can be inherited, a phenomenon known as transgenerational epigenetic inheritance (TEI (Miska and Ferguson-Smith, 2016)). Epigenetic change affects the way genes are regulated and expressed, without altering DNA sequence (Heard and Martienssen, 2014). Although these marks are generally erased during gamete formation and early embryonic development (Nashun et al, 2015), some of the epigenetic signals can survive this type of reprogramming and pass across generations, by which acquired traits and environmental effects can be transmitted to the progeny (Houri-Zeevi et al, 2021).

Darwinian evolution requires heritable variation in the population. Mutation at the DNA sequence level generates different genotypes within the population. These genotypes then change in their frequency over time within the population, due to differential survival and reproduction of different genotypes (natural selection) or random effects due to finite population sizes (genetic drift)

(Blount et al, 2018). The existence of transgenerational epigenetic inheritance suggests that epigenetic variation might contribute to evolution in the same way (Ashe et al, 2021; Stajic and Jansen, 2021).

It seems highly likely that individual cases of evolution being driven directly by epigenetic changes will exist in biology. Cancer cells, which display mitotic inheritance in which epigenetic information is often stably transmitted (see below), undoubtedly evolve due to epigenetic changes that are heritable (Shlyakhtina et al, 2021). Moreover, in organisms under laboratory conditions, several examples of this have already been demonstrated (Pérez-Arques et al, 2025; Calo et al, 2014; Stajic et al, 2019; Torres-Garcia et al, 2020). However, the focus of this review is to assess whether these kinds of effects are unusual "edge cases" or hint at a widespread role for epigenetic changes as drivers of evolution that qualitatively alter the rate or direction of evolution (Sabarís et al, 2023).

In this review, we focus on a key aspect of the ability of epigenetic changes to drive evolution: epimutations. Epimutations are variations in epigenetic state that can appear within populations. Just as understanding DNA sequence variation (mutations) is central to understanding classical Darwinian evolution, understanding epimutations is key to understanding the role of epigenetics in evolution. We first review the key mechanistic principles of transgenerational epigenetic inheritance, as this explains the fundamental bases for how epimutations arise. We then discuss the extent to which epimutations appear within populations and how stably they are propagated. This enables us to explore the extent to which natural selection and genetic drift could act on epimutations without DNA sequence variation. Finally, we discuss how epimutations might synergize with classical DNA sequence mutations to shape evolutionary processes. Our main focus will be a discussion of these processes in animals; however, examples from plants and fungi will be used to illustrate key theoretical principles.

## Transgenerational epigenetic inheritance: the prerequisite for epimutations

Epigenetic information is not directly encoded in the DNA sequence but can be stably propagated through cell division (Holliday, 2006). The classic example of this is in development: each cell in a multicellular organism has an identical genome, but different cell types have very different programs of gene expression, due to differences in epigenetic regulation. Importantly, epigenetic differences are heritable through mitosis: gene expression programs remain stable through cell division. In order for this to happen,

Department of Biochemistry, University of Oxford, Oxford, UK. ✉E-mail: peter.sarkies@bioch.ox.ac.uk

mechanisms exist that propagate epigenetic variation through cell division (Kaufman and Rando, 2010). Many different types of epigenetic information exist, including histone modifications (Bannister et al, 2001), covalent modifications to DNA (mostly 5-methyl-cytosine) (Bird, 2002), transcription factors that bind DNA and promote their own synthesis (Harvey et al, 2018) and small noncoding RNAs that either directly or indirectly promote synthesis of new small RNAs of the same type (Cecere, 2021). Just because a molecular mechanism can be epigenetic, it does not mean that it acts in this way under all circumstances, and different cell types rely on different epigenetic mechanisms (Sarkies and Sale, 2012). Moreover, often several different epigenetic mechanisms combine to regulate the same locus, with feedback between them making the epigenetic state more robust.

Epigenetic information can also be transgenerationally inherited. Importantly, what is meant by "transgenerational" is nuanced. In general, transmission between an organism (P0) and its direct progeny (F1) is not sufficient to be defined as transgenerational because the germ cells are exposed to the same environment as the P0 organism. Transmission through a single generation is therefore known as "intergenerational" and encompasses a range of mechanisms such as maternal provision of proteins that define the environment of the early zygote (Perez and Lehner, 2019). Transmission of epigenetic information through at least F2 is required for a phenomenon to be labeled as transgenerational. Indeed, in female mammals, transmission through two generations is also "intergenerational" because the germ cells of the F1 organism are also exposed to the same environment as the P0 germ cells, so F3 is required in this case (Miska and Ferguson-Smith, 2016). Mechanistically, transgenerational epigenetic inheritance to F2 and beyond must require some form of reamplification of the original epigenetic signal (Fitz-James and Cavalli, 2022).

The propensity of epigenetic mechanisms to act transgenerationally can be evaluated in the same way as epigenetic transmission through mitosis. There must be a mechanism to enable stable propagation of the modification across generations. This does not exist for all modification, even those that are stably transmitted through mitosis in development. A good example illustrating this point is cytosine DNA methylation. In the CG sequence context, 5-methylcytosine (5mC) can be transmitted epigenetically through cell division. This is because of an enzyme, maintenance methyltransferase (DNMT1 in animals), that recognizes hemimethylated DNA that forms during replication of DNA containing 5mC (Song et al, 2011). It then introduces methylation onto the other strand, thus perpetuating the original modification state (Holliday et al, 1987). In principle, therefore, altered methylation could be transmitted transgenerationally. This occurs in plants, where epigenetic variation due to DNA methylation can be stably transmitted and result in phenotypic differences (Law and Jacobsen, 2010). However, in mammals, this is unlikely because most methylation is removed from the genome during the development of germ cells and again post-fertilization in the early embryo (Nashun et al, 2015). This poses a barrier for transgenerational epigenetic inheritance of DNA methylation. Indeed, even at regions of the genome that escape erasure. Experimental induction of DNA methylation using epigenome editing can result in transmission over several generations. However, natural examples of mammalian transgenerational epigenetic inheritance driven by DNA methylation are likely to be unusual (Anastasiya Kazachenka

et al, 2018). It is important to note that the wholescale eradication of DNA methylation in mammals is not widely conserved (Bogdanović et al, 2016) and likely evolved as a specific process to facilitate imprinting rather than development in general (de Mendoza et al, 2020). It remains possible therefore that transgenerational epigenetic inheritance due to DNA methylation could be found in some animals, but this has yet to be fully investigated at the molecular level.

Another route for transgenerational epigenetic inheritance is post-translational modifications of histone proteins. Histone modification patterns are stably inherited through mitosis due to mechanisms that introduce modifications into newly synthesized histones guided by surrounding parental nucleosomes (Reverón-Gómez et al, 2018). To give a concrete example of this, H3K27me3 domains are inherited because the methyltransferase that introduces H3K27me3 (PRC2) contains a domain that recognizes H3K27me3. Thus, newly synthesized histones acquire H3K27me3 modifications if they are deposited alongside parental nucleosomes containing H3K27me3 during replication. These mechanisms are robust through mitosis; however, transgenerational transmission of histone modifications is more challenging because in most animal species, histone proteins are removed genome-wide during the development of sperm and replaced with protamines (Balhorn, 2007). Nevertheless, there is potential for some histone post-translational modifications to be transmitted through sperm (Gaspa-Toneu and Peters, 2023). Indeed, in C. elegans, some histone post-translational modification states have been shown to be transmitted through sperm (Tabuchi et al, 2018). Protamine replacement does not happen in oocytes. In Drosophila and C. elegans, histone modification changes can be transmitted into embryos via this route (Gaydos et al, 2014; Atinbayeva et al, 2024). There is also evidence for transgenerationally inherited changes in chromatin state that persist for several generations in both organisms (Fitz-James and Cavalli, 2022; Özdemir et al, 2025).

The best-understood mechanism of transgenerational epigenetic inheritance in animals is small noncoding RNAs in C. elegans (Cecere, 2021). C. elegans possesses RNA-dependent RNA polymerases (RdRPs) that produce small noncoding RNAs using the target RNA as the template (Pak and Fire, 2007). RdRPs produce a specific type of small RNAs that are 22 nucleotides long and start with a G (22Gs; see Fig. 1). 22G-RNAs bind to Argonaute proteins that stabilize them and also enable recruitment of other complexes to enact transcriptional and post-transcriptional silencing (Cecere, 2021). Crucially, RdRPs participate in a positive feedback loop because small RNAs that recognize the target through complementary base pairing recruit RdRPs to the target, thus initiating the formation of more small RNAs (Sapetschnig et al, 2015). In this way, transmission of a few small RNAs transgenerationally can result in stable propagation because RdRP activity will replenish the pool in the next generation. Silencing of transgenes initiated by RNAi or specific small RNAs can in this way be inherited for many generations (Ashe et al, 2012; Luteijn et al, 2012; Shirayama et al, 2012). Importantly, this mechanism of inheritance is not dependent either on the DNA itself or chromatin and can be shown to be transmitted through the cytoplasm (Rieger et al, 2023). The exact requirements for Argonaute proteins in the P0 and F1 generation is not fully established, but it appears that some Argonautes are specific for establishing new silencing responses, while others transmit the 22G-RNAs between generations (Schreier et al, 2025; Woodhouse et al, 2025).

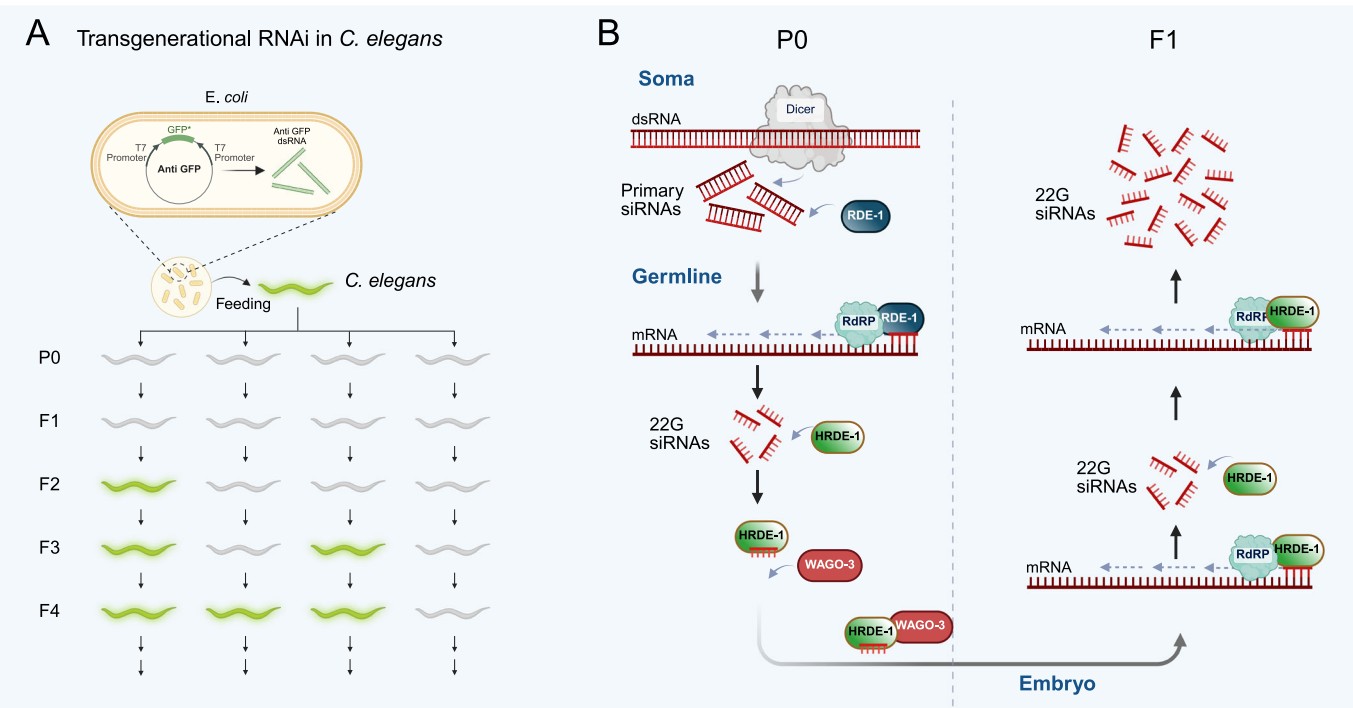

**Figure 1. Summary of transgenerational epigenetic inheritance in *C. elegans*.**

(A) Feeding *C. elegans* with bacteria expressing dsRNA matching to GFP leads to silencing of the GFP that is inherited, potentially for many generations, independent of the initial trigger. (B) Current understanding of the mechanism, which depends on Argonaute proteins that bind small RNAs and transmit them between the generations, and RNA-dependent RNA polymerases (RdRP) that regenerate small silencing RNAs in the next generation, leading to the continuation of silencing.

The mechanism of transgenerational epigenetic inheritance in *C. elegans* serves to emphasize the key mechanistic requirement for non-genetic information to be passed on between generations: some form of amplification mechanism that can prevent dilution between successive generations (Houri-Zeevi et al, 2020). For small RNAs, this role is fulfilled by RdRP, an enzyme which is not present in mammals where there is no RdRP, thus making it unlikely that small RNAs act transgenerationally. RdRPs do exist in many other animals (Lewis et al, 2018; Zong et al, 2009), however, whether these could contribute to transgenerational epigenetic inheritance in other organisms besides nematodes is unknown.

Just as distinct epigenetic mechanisms tend to combine in gene regulation during development, transgenerational epigenetic inheritance mechanisms may also incorporate several molecular pathways simultaneously. Small RNA-mediated epigenetic inheritance in *C. elegans* also requires histone modifications at some loci, in particular H3K9 methylation (Ashe et al, 2012) and H3K23 methylation (Schwartz-Orbach et al, 2020). Nevertheless, it is clear that small noncoding RNAs rather than chromatin are the primary molecules that transmit epigenetic memory between generations (Woodhouse et al, 2018, 2025).

## Properties of epimutations in populations

The existence of robust mechanisms to transmit some types of epigenetic inheritance is the first requirement for the generation of epimutations. Mechanistic studies of transgenerational epigenetic inheritance, however, have been performed with reporter genes or specific individual loci. To understand how epimutations might contribute to the processes of evolution, it is essential to evaluate genome-wide properties of epimutations. A natural reference point for comparison is classical DNA sequence mutations: how do epimutations compare to genetic mutations? In this section, therefore, we will discuss how the rate, stability, genome-wide distribution, and effects of epimutation compare to what is known about DNA sequence mutations.

To evaluate the properties of epimutations, it is crucial to use a framework in which the epimutation can be studied independently of its effect on fitness. Without this, epimutations that have a negative impact on fitness would be eliminated from the population by purifying selection and thus underestimated in the epimutations that are observed. To minimize the extent of purifying selection, an approach can be borrowed from traditional evolutionary biology known as Mutation Accumulation (MA). In the MA framework, multiple lineages are propagated independently from a common ancestral population (Katju and Bergthorsson, 2019). These populations are passed through a series of bottlenecks, ideally minimizing the population to only one individual (if the organism is hermaphrodite or parthenogenic) or one breeding pair. Under these conditions, genetic drift is responsible for most changes in allele frequencies that occur in the population (Katju and Bergthorsson, 2019). Any mutations that arise therefore have an equal probability to go to fixation regardless of whether they are beneficial, neutral, or deleterious (Lynch and Conery, 2003). To estimate the rate of mutations, the genomes of the populations are

sequenced after several generations and compared to the ancestral population. Exactly the same approach can be employed to study epimutations, which we term eMA for epimutation accumulation (van der Graaf et al, 2015).

## Defining an epimutation

Just as there are many distinct types of mutation, such as single-nucleotide variants (SNVs, or point mutations), structural rearrangements, and copy number variants, epimutations could potentially come in many different forms. The simplest type of epimutation is typified by cytosine methylation. Methylation at a cytosine could either appear or disappear in an individual, and that change could be passed on to its descendants (Holliday et al, 1987). The cytosine thus has two distinct states, and conversion between them is an epimutation. However, methylation sites across the genome tend not to behave in isolation, so it is also useful to consider the methylation of many cytosines within a defined region, such as a gene or a promoter (van der Graaf et al, 2015). This gives rise to a potential complexity: how to define what counts as an epimutation? Even if each cytosine can only adopt two states, if there are many cytosines within a region, there may be many different possible methylation levels. Crucially though, it turns out that regions tend to show bimodal distributions in methylation: either most cytosines are methylated or very few are methylated (Denkena et al, 2021). As a result, larger-scale epimutations can be defined as "differentially methylated regions (DMRs)" with two states, just as individual cytosines (Denkena et al, 2021).

Epimutations due to histone post-translational modifications are also likely to show bimodality, as genomes tend to have broad domains containing high levels of a few histone modifications and low levels of the others (Carelli et al, 2017). Mathematical modeling of post-translational histone modification states indicates bimodality (Briffa et al, 2024; Movilla Miangolarra and Howard, 2025). Supporting this, cases of transgenerational epigenetic inheritance involving histone modifications similarly display switch-like behavior between different modification states (Fitz-James et al, 2025).

Small noncoding RNAs in *C. elegans* also display bimodality, as their formation depends on RNA-dependent RNA polymerase. Because this enzyme can act at many sites across a target RNA, each time synthesizing a distinct 22G-RNA antisense to the target gene, it means that once targeted, levels of small RNAs build up rapidly (Pak and Fire, 2007). As a result, small RNAs mapping to a target are bimodal: either very few or many small RNAs (Beltran et al, 2020). This can be used to define an epimutation by using a two-state model similar to that employed for DNA methylation (Beltran et al, 2020).

These mechanistic considerations mean that most studies adopt a two-state model for defining epimutations, and the following discussion will adhere to this. However, it is worth noting that more subtle quantitative variation in epigenetic states may be possible at some loci, which would make estimation of rates of epimutation significantly more complicated.

## The rate and stability of epimutations

The first experiments employing eMA lines to investigate epimutations used the model plant *Arabidopsis thaliana*. Epimutations were assayed by bisulfite sequencing to identify changes in cytosine methylation that accumulated in the different populations (Schmitz et al, 2011; Becker et al, 2011). In plants, methylation can take place at CG, CHG, and CHH sequence contexts (where H indicates A, C, or T), and all three showed much more rapid divergence compared to the DNA sequence, indicating epimutation rate was 10–100-fold greater than the DNA mutation rate. Careful statistical analysis of many such experiments allowed estimation of both "forward" (gain of cytosine methylation) and "back" (loss of cytosine methylation) epimutation rate, with an estimate of around 1e-4/cytosine/generation for both (van der Graaf et al, 2015). Importantly, similar results were obtained whether individual cytosines or regions encompassing many cytosines (DMRs) were considered. Indeed, DMRs based on CG methylation show very similar rates of epimutation as individual CG show a strikingly similar overall epimutation rate to individual CGs (Denkena et al, 2021). CHH and CHG DMRs have ~100-fold lower rates of epimutation (Denkena et al, 2021).

A consequence of the rapid forward and back epimutation rate is that epimutations are unstable over large numbers of generations. This can be seen clearly from a comparison to DNA sequence mutations, which are usually treated as "stable" (i.e., irreversible) in the modeling of evolution (Charlesworth and Jain, 2014). DNA mutations can only be approximated as stable because the probability of a mutation occurring at exactly the same nucleotide is very low: mutation rates for a typical eukaryote are of the order of 1 per $10^8$ base pairs per generation (Lynch et al, 2016), and there are 4 bases so the probability of exact reversal is $2.5*10^{-9}$. Epimutation rates are much higher, and there are only two states (methylated and unmethylated) (van der Graaf et al, 2015; Johannes, 2024). Therefore, the expected number of generations that an epimutation lasts for is much shorter than the expected duration of a mutation.

In animals, epimutation rates have also been measured using eMA lines in the nematode *C. elegans* (Beltran et al, 2020; Wilson et al, 2023; Fallet et al, 2023). Using small noncoding RNA sequencing, epimutations were defined as the accumulation of small RNAs mapping to a specific gene (see above). A median of around 100 genes showed significant alterations in small RNAs mapping to them in each generation, while approximately one gene per generation acquires a point mutation in analysis of *C. elegans* MA lines (Katju and Bergthorsson, 2019), thus the rate is considerably higher for epimutation than DNA sequence mutations (Beltran et al, 2020; Fallet et al, 2023). The median duration of epimutations was shown to be around 3–4 generations, thus indicating that small RNA-mediated epimutations are short-lived just as DNA methylation epimutations in Arabidopsis. Further studies incorporated ATAC-seq to correlate changes in small RNAs to changes in chromatin environment. Changes in chromatin accessibility were observed to occur at a similar rate to small RNA-mediated epimutations (Wilson et al, 2023). In some cases, these overlapped, and this was associated with longer duration of epimutation, suggesting that cross-talk between small RNAs and chromatin (Woodhouse et al, 2018) could act to stabilize epimutations.

An important implication of the rapid rate and short duration of epimutations is that divergence at the epigenetic level will be expected to be very different from divergence at the DNA sequence level. Because DNA mutations are rare and very stable, mutations accumulate in a linear fashion, both in MA lines and in separate populations. This is what gives rise to the hypothesis of the

molecular clock used to estimate when different lineages separated. Epimutations, however, will cause rapid initial divergence, but this rate will quickly tail off, because an equilibrium between new epimutations and the reversal of old ones will be reached (Yao et al, 2021). In the *C. elegans* epimutation experiments, divergence in small RNAs was clearly apparent after 2–4 generations, but the number of differences to the ancestral population was the same after 75 generations as after 25 generations, indicating saturation had been reached (Beltran et al, 2020). Nevertheless, it is still possible to use epimutations as a short-term molecular clock with the potential to have closer resolution than DNA sequence-based clocks, as long as the reversibility is taken into account in the mathematical formulations (Yao et al, 2021).

One important caveat to these studies of epimutation properties is the heterogeneity of epimutations within different samples. In *C. elegans*, as it is not currently possible to sequence small noncoding RNAs from a single worm, the level of small RNAs was estimated from a population all descended from one individual, so the epimutation state was inferred from the average level of small noncoding RNAs across the population. Loss or gain of an epimutation may therefore represent loss of the epimutation in a sufficient number of individual offspring rather than loss of the epimutation in the parent. Examining epimutations at the single worm level is a really important next step for the field.

## The genome-wide landscape of epimutations

In addition to the occurrence and duration of epimutations, another fundamental property to evaluate is their genome-wide distribution. Evaluation of eMA lines can be used to estimate the probability that any particular gene acquires an epimutation. These analyses show that some genes are more likely to epimutate than others. In *C. elegans*, genes that are prone to epimutation tend to be targets of a specific type of small noncoding RNAs known as piRNAs (Piwi-interacting small RNAs) (Beltran et al, 2020). Moreover, they are enriched for genes that show increased inter-individual variability in expression, whereas stably expressed "housekeeping" genes are less likely to acquire epimutations. Epimutable genes also tend to be enriched for genes with roles in xenobiotic defense (Wilson et al, 2023).

On the basis of these results, it is tempting to speculate that the propensity of certain genes to epimutate might be advantageous at the organismal level. For example, if some genes involved in xenobiotic defense can get switched on randomly in a population, it might mean that some individuals and their descendants are resistant to exposure to a novel toxin that enters the environment (O'Dea et al, 2016). This would imply that natural selection acts to shape the mechanisms that give rise to epimutations such that they are more prominent at certain regions and suppressed at others (such as housekeeping genes). Intriguingly, this idea connects to recent debates concerning traditional DNA sequence mutations. DNA sequence mutations are known to occur non-randomly across the genome (Lynch et al, 2016). Recent analyses of mutations in *Arabidopsis* suggested that genes with essential functions were protected from mutations and proposed that this was the result of natural selection (Monroe et al, 2022). Indeed, a mechanism was proposed to account for this due to the activity of a DNA repair enzyme being directed to housekeeping genes by their distinct chromatin structure (Quiroz et al, 2024; Monroe et al, 2022). On the other hand, it is possible that the genome-wide distribution of epimutations is due to mechanistic aspects that affect the propensity of individual genes to switch their epigenetic state and not shaped by natural selection.

## The effects of epimutations

A crucial question in population genetics concerning classical DNA sequence mutations is their expected effect on fitness (Eyre-Walker and Keightley, 2007). This concept can be understood with the help of an idealized experiment. Take a mutation that has arisen at random in a population, and engineer it into the organism in isolation and compare the fitness of this organism compared with the starting genotype. Collating the effects of multiple mutations would result in a distribution (Fig. 2A), known as the distribution of fitness effects (Loewe and Hill, 2010). Note that this is not the entire spectrum of possible mutations, such as would be obtained by saturation mutagenesis (deep mutational scanning, see for example (Somermeyer et al, 2022)), because not all mutations are equally likely due to biases in mutation rate genome-wide(Lynch et al, 2016). Except in limited cases, such as viruses, actually performing this experiment for an organism is currently impossible. However, much effort has gone into inferring the distribution of fitness effects from experiments such as mutation accumulation experiments (Crombie et al, 2024) and careful analysis of population genetics data (Keightley and Eyre-Walker, 2010). The general consensus is that most mutations have a small fitness effect, which is more likely to be detrimental than beneficial (Huber et al, 2017; Crombie et al, 2024). Due to limitations in the type of mutations that can be detected through mutation accumulation and population genetics approaches it is possible that there is a bimodal distribution such that there is an enrichment for mutations with a very severe effect (Fig. 2A).

The distribution of fitness effects of epimutations has not been formally studied. However, we can put forward some hypotheses. In *C. elegans*, fitness decline during MA experiments can be used to estimate the mean effect of a mutation (Crombie et al, 2024; Katju et al, 2015). The rate of fitness decline is slow, consistent with the conclusion that the mean effect of a mutation is very weakly deleterious, or that strongly deleterious mutations are rare (Crombie et al, 2024). In one experiment, for example, after ~400 generations, the fitness had declined by ~40% (Katju et al, 2015). After one generation, therefore, fitness decline is going to be far too small to detect (Fig. 2B). After 400 generations of MA, there were ~200 genetic mutations per line (Konrad et al, 2019). Crucially, there are >100 new epimutations each generation (Beltran et al, 2020). If the effect of an epimutation were similar to a genetic mutation then a fitness decline would be apparent after only a very small number of generations, which has not been observed (Fig. 2B). We can conclude from this that the majority of epimutations are likely to have a very small effect on fitness, or equivalently that epimutations with a large deleterious effect are very unlikely.

This conclusion might intuitively seem surprising because epimutations, at least as defined in *C. elegans*, involve the gain or loss of small noncoding RNAs across the entire gene, which is often associated with a significant change in expression (Beltran et al, 2020). Why are these even less likely to affect fitness than genetic mutations, which are often synonymous or intergenic? One

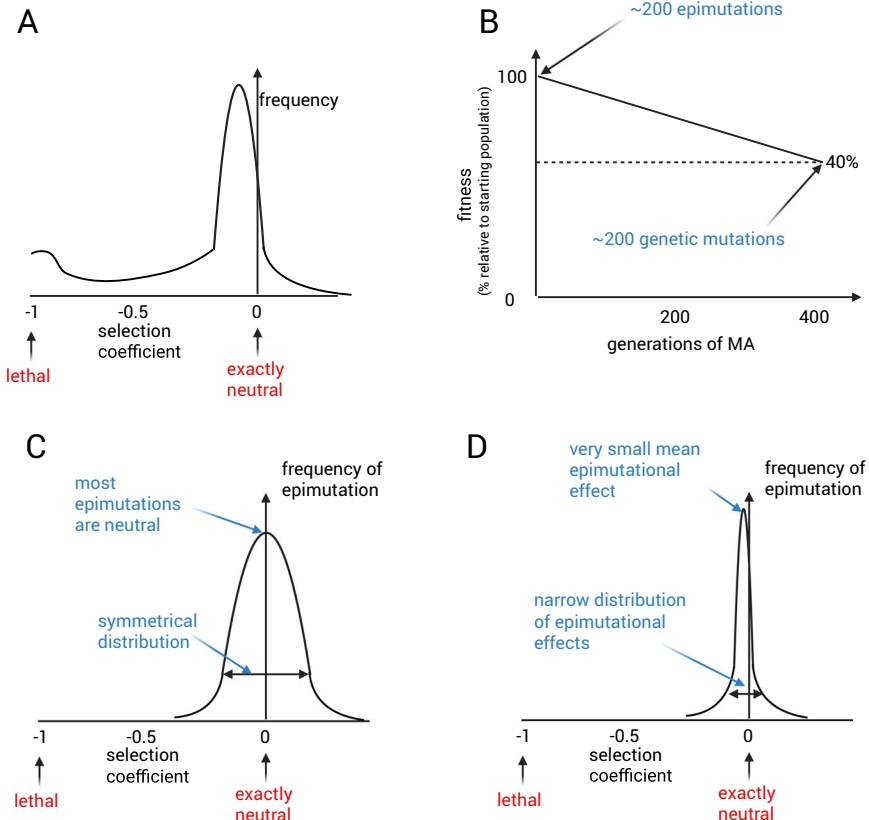

**Figure 2. Distribution of fitness effects for mutations and epimutations.**

(A) shows the consensus view of the distribution of fitness effects for new DNA sequence mutations. (B) shows a hypothetical graph showing fitness decline over 400 generations of mutation accumulation. Fitness decline after 400 generations is taken from (Katju et al, 2015). (C, D) show different hypothetical distribution of fitness effects for epimutations, with either a symmetrical broad distribution or a very narrow distribution centered around a point close to neutral for the selection coefficient.

possibility is that epimutations are equally likely to be beneficial or deleterious, resulting in a symmetrical distribution of fitness effects. Since there are so many epimutations, the expected total effect of epimutations would therefore sum to ~0 (Fig. 2C). This would imply that it is much more likely for an epimutation to have beneficial effects compared to a mutation, which seems unlikely, but is certainly possible. Alternatively, the biased genome-wide distribution of epimutations such that they are extremely strongly depleted from important genes might lead to a very narrow distribution of fitness effects compared to genetic mutations (Fig. 2D). This would imply that selection has shaped the genome-wide distribution of epimutations, which is certainly plausible because epimutations are common so there would be a strong fitness benefit to preventing them from occurring at housekeeping genes.

An interesting possible way to investigate whether epimutations do have a small negative effect on fitness would be to track fitness decline in MA lines over much shorter numbers of generations. If epimutations have a small but significant negative effect on average, then there would be an initial decline at a faster rate, which would then slow down as the divergence saturates. If, instead, the fitness decline follows the same trajectory across the generations or even speeds up at later generations due to epistatic effects between accumulating DNA sequence mutations (Johnson et al, 2023), then

this would suggest that epimutations on average are neutral with equal beneficial and detrimental probability.

# Epimutation and adaptation by natural selection

eMA experiments in both *Arabidopsis* and *C. elegans* have convincingly shown that epimutations occur within populations under conditions where drift predominates. Despite the fact that large-effect epimutations are probably rare, some epimutations could contribute to fitness and thus could contribute to adaptation in large population sizes where natural selection operates. However, their short duration of <10 generations makes it hard to see how they could contribute to natural selection (Webster and Phillips, 2024; Charlesworth and Jain, 2014). The fundamental reason for this is the heritability of epimutations. Heritability can be estimated from the median duration of an epimutation (see Box 1, Fig. 1). The expected duration of epimutation in the absence of selection drops off very rapidly with even a small decrease in heritability. Using this approach, an epimutation that has a median survival of four generations in a *C. elegans* population has a heritability of about 0.80; in other words about 20% of the offspring of any individual worm will not carry the epimutation (sometimes expressed as

"reversal probability", given the symbol β). This makes it more difficult for epimutations to influence evolution because the selection coefficient (which measures how "beneficial" an epimutation or mutation is) has to be higher for any given epimutation to reach high frequency than for a DNA sequence mutation, which has much higher heritability (see Box 1 for a more detailed explanation of this point).

Despite this, there is a possibility that a beneficial epimutation could drive directional evolution by natural selection if the selection coefficient is very strong (Webster and Phillips, 2024). This can be seen by considering the example of a *C. elegans* epimutation with a heritability of 80%. The fact that 20% of offspring of any individual lack the epimutation means that if it has no effect on fitness, it would be lost from the population rapidly, hence the short median duration. However, if the majority of the 20% of offspring that do not have the epimutation die or fail to reproduce, then the epimutation could be maintained and, if the benefit of the epimutation is very strong indeed, the frequency of the epimutation can even approach 100%, analogous to fixation for a genetic change (see Box 1, Fig. 2).

A concrete demonstration of this idea comes from experiments in the fungus *S. pombe*. A screen to detect potential epimutations that provided resistance to caffeine was specifically designed to select for resistant isolates where the resistance was dependent on the continued presence of caffeine in the environment (Torres-Garcia et al, 2020; Yaseen et al, 2022). Crucially, this does not imply that the epimutation is directly promoted by caffeine. Instead, the epimutation is lost stochastically with high probability each generation, so without selection, the epimutation is rapidly lost from the population. Similarly, in plants, strong selection over a short time was shown to result in consistent epigenetic differences in loci associated with flowering time (Schmid et al, 2018), These examples indicate that it is at least possible that epimutations might confer strong enough effects to overcome their limited heritability, and contribute to adaptation.

The question remains open, however, as to whether this could meaningfully affect evolution by natural selection in animals in the environment. Situations where the fitness effect of the epimutation is large enough for it to overcome its low heritability are likely to be rare. Moreover, as soon as the selection pressure was reduced the epimutation would rapidly disappear. Long-term separation of populations, such as required for speciation, for example, seems unlikely to be due to epimutations alone. Even over the very short-term, genetic variation existing in a natural population is likely to overwhelm epigenetic variation in driving adaptation (Schmid et al, 2025).

## Epimutations as facilitators of genetic change

Epimutations alone may have limited potential to drive evolution by natural selection. Nevertheless, the interplay between epimutations and DNA sequence mutations could indicate a role for epimutation in long-term evolutionary processes (Webster and Phillips, 2024). Crucially, epimutations occur much more frequently than mutations. As a result, they are more likely to arise, so would be expected to occur earlier in a population exposed to a novel stress. As discussed above, the epimutation could be maintained within the population at a high frequency if it is

strongly beneficial. Eventually, however, a genetic mutation with the same effect as the epimutation may arise (Sarkies, 2020). A genetic mutation with the same effect as an epimutation can be maintained at a higher frequency within the population655455. As a result, the genetic mutation would be at an advantage and would replace the epimutation (see Box 1, Fig. 3).

It is important to emphasize that epimutations can make genetic mutations more likely at the population level without having a direct effect on the probability of mutation itself. The reason for this is that if the epimutation increases fitness, the population size will be increased if it occurs and is inherited, even if the heritability is low (Webster and Phillips, 2024). As a result, even if mutations occur at a constant rate per genome per generation, the time expected for a beneficial mutation to arise would be shorter because there are more individuals within which the mutation could occur. Thus, the presence of an epimutation would be expected to accelerate the process of genetic change within the population.

Even though no direct effect of epimutation on mutation rate is necessary for epimutation to accelerate evolution, certain types of epimutation may indeed accelerate local mutation rate, thus lead to even faster rates of genetic adaptation. The rate of mutation at any given region of the genome can be predicted very accurately by many factors that can act epigenetically, most notably CpG methylation and repressive histone modifications (Schuster-Böckler and Lehner, 2012; Supek and Lehner, 2015). These correlations have been most clearly defined in somatic cells, but they also apply to germline mutations in mammals (Rahbari et al, 2016) and, recently, an association between active chromatin and reduced mutation rate has been shown to operate at the population level in *Arabidopsis* as discussed above (Monroe et al, 2022; Quiroz et al, 2024). Mechanistically, the origin of these correlations is multi-faceted, involving changes in replication timing and direct influences on the rate of DNA repair (Supek and Lehner, 2019). Though an association between small RNA-mediated epigenetic change and mutation rate has yet to be tested; gain of small RNA-mediated silencing at a locus could lead to increased mutation rate due to repressive histone modifications that often accompany small RNAs (Cecere, 2021). Interestingly, this argument would suggest that epimutations that lead to silencing (gain of CpG methylation at promoters; repressive histone modifications) might accelerate genetic change more than epimutations that relieve silencing, and therefore might have a more important role in evolution within populations.

In addition to influencing the local mutation rate, epimutations might also influence mutation rate genome-wide through the activity of transposable elements. Transposable elements are mobile elements within the genome that can jump into new sites across the genome. Epigenetic mechanisms keep transposable elements silenced; thus, epimutations that reduce silencing of a specific transposable element locus could lead to the reactivation of transposons and integration into other regions of the genome. Evidence for altered transposable element insertions due to epimutations has come from experiments in plants, showing that epimutations changing DNA methylation states led to differences in transposable element mobilization (Quadrana et al, 2019). Increasing genome-wide mutation in general would not normally be expected to promote adaptation because deleterious mutations are more likely than beneficial ones. However, transposable element insertions are much more likely at certain genes, particularly those

### Box 1   Population genetics and epimutations: the Sisyphean problem

Population genetics aims to understand and predict the fate of different variants (alleles) in populations. One of the most important concepts in population genetics is fixation. An allele is said to go to fixation when all the individuals in the population are homozygous for the allele. Fixation of alleles is central to divergence between populations and ultimately the generation of distinct species. A key question, therefore, that population genetics aims to answer, is how likely it is that a new allele that enters the population by mutation will go to fixation. Two major effects determine whether alleles go to fixation: natural selection and genetic drift. Natural selection acts to promote fixation of alleles that increase fitness, because individuals with the allele have greater reproductive success. The fitness of a new mutant allele relative to the original genotype is referred to as s, such that positive s is beneficial, negative s is detrimental and $s = 0$ means the allele is neutral. Alleles with large, positive, s, will generally go to fixation rapidly, while alleles with negative s will be eliminated. Genetic drift is a random process which operates because the number of individuals in any population is not infinite so the genetic makeup of the population may change from generation to generation due to taking only a subsample of the total genetic diversity. Genetic drift is stronger when the number of individuals that can reproduce in the population (effective population size, Ne) is small. If genetic drift alone is operational, the selection coefficient makes no difference to the probability of fixation of a new mutation- beneficial, neutral, and detrimental alleles are all equally likely to become fixed.

Putting this together, population genetics equations show that the fate of a new mutant allele in the population over evolutionary time can be predicted from two factors: effective population size $N_e$ and the fitness coefficient s:

1) $N_e$ s»1 means that it is very likely that a beneficial allele will go to fixation
2) $N_e$ s«1 means that genetic drift is operational, so beneficial, detrimental, and neutral alleles are all equally likely to go to fixation.

The competition between drift and selection is often referred to as the "drift barrier" (Lynch and Conery, 2003). Essentially, if the population size is very small, the selection coefficient has to be much larger for $N_e$s to be above the drift barrier, whereas with a large population size, a smaller selection coefficient is needed. Focusing specifically on beneficial mutations, only very beneficial mutations will overcome the drift barrier in a small population ($N_e$ <100).

In the case of an epimutation, however, there is an additional barrier beyond the drift barrier which is imposed due to the reduced heritability of the epimutation. For a genetic mutation, reversal is rare so all offspring of an individual will most likely contain the genetic mutation. An epimutation, though, carries a substantial probability of reversion, as evidenced by their short half-life within populations (see Box Fig. 1). An epimutation therefore has to be even more beneficial because it has to overcome both the drift barrier and the low heritability. We call this the Sisyphean problem after the mythic figure Sysyphus who was condemned to role a stone up a slope every day only for it to fall back down at night. Even epimutations with a relatively strong beneficial effect s ~ 0.1 are unlikely to ever reach anything approaching fixation (Box Fig. 2). Instead, they reach an equilibrium where the forward and back rate are balanced (Yao et al, 2021), and the position of equilibrium depends on the magnitude of s (Box Fig. 2).

Comparison to genetic mutations is interesting, however. Genetic mutations of the same fitness coefficient as epimutations will be much more likely to reach very high frequencies and go to fixation. However, this happens a lot more slowly (Box Fig. 3). The reason for this is that when a new mutation arrives in the population, it has only one shot: mutations are rare, so the chance of the same mutation arising randomly again is low unless the population is huge. Epimutations, however, are much more frequent so the loss of epimutations due to drift and low heritability can to some extent be compensated for by frequent reintroduction through new epimutations arising independently. This is why an equilibrium is reached quickly, over 10 or so generations in the simulation illustrated here, whilst genetic mutation takes ~100 generations.

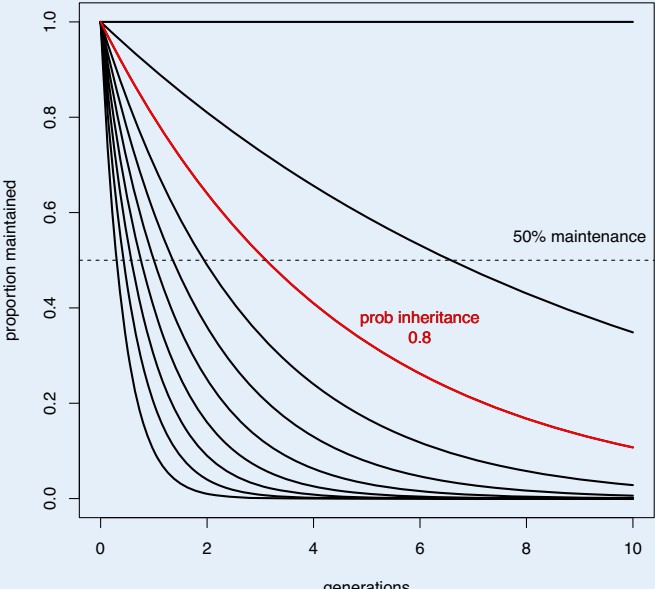

**Box Fig. 1 Maintenance of an epimutation in the population at different levels of heritability (the percentage of offspring that maintain the epimutation).** Calculation based on the model used in (Webster and Phillips, 2024).

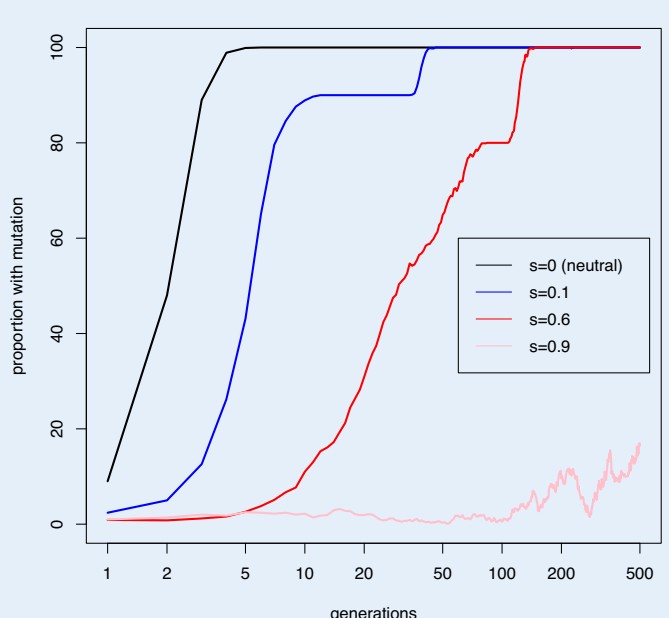

**Box Figure. 2 Fate of a new beneficial epimutation.** Simulation of how the frequency of an epimutation across a range of selection coefficients changes over time. Epimutations were introduced at a frequency of $1/N$ where $N$ is the population size. $N$ was set to 100 for all simulations. Each trace represents the average of 10 separate simulations. The epimutation rate was 20% (i.e., a heritability of 80%). Note that the selection coefficients where the epimutation approaches high frequency in the population are extremely high, much greater than is the case for genetic mutations, where $s = 0.1$ is considered a very strong selection coefficient. Source data are available online for this figure.

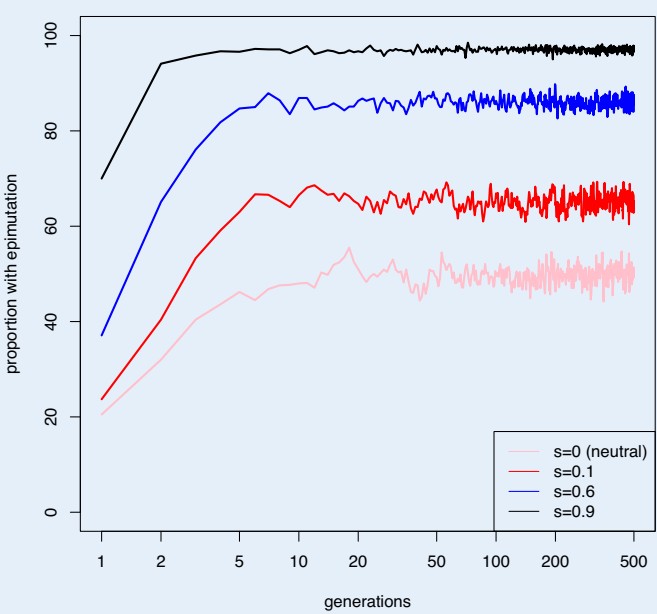

**Box Figure. 3 Fate of a new beneficial mutation.** Simulation of how the frequency of a mutation across a range of selection coefficients changes over time. The parameters of the simulation were exactly the same as for the epimutation with the exception that the mutation rate was set at 100-fold lower than the epimutation rate. Compare to the fate of a new epimutation in Box Fig. 2. Source data are available online for this figure.

that are environmentally responsive, so the genome-wide increase in transposon mobilization may promote adaptive mutations.

Once a mutation with the same or similar effect to an epimutation has arisen, the epimutation may become rapidly lost from the population. This is because if the genetic change has the same effect as the epimutation, the fitness benefit of having both the genetic change and the epimutation would be expected to be smaller than expected from the fitness effect of each individually, in other words the epimutation and the genetic change would be epistatic. A simple example of this would be to consider the fission

yeast caffeine resistance phenotype described above. At the molecular level, this occurs due to an epimutation which silences the gene *hba1* the locus (Torres-Garcia et al, 2020). A point mutation such as a stop codon within *hba1* can also give rise to resistance (Castillo et al, 2003), and in this background the silencing would not provide any additional resistance. Due to the low heritability of the epimutation, if there is no selection, it would rapidly be lost through drift. Of course, this presupposes that epimutation and genetic change at the same locus would show epistasis. Whilst plausible, this is not necessarily true and would be an interesting hypothesis to test, perhaps using deep mutational scanning combined with epigenetic editing.

Assuming that epimutations and genetic mutations at the same locus tend to be epistatic, evidence of epimutations would be difficult to obtain from a comparison of different populations. Epimutations would occur early in the adaptive process, but would be lost after genetic change becomes fixed (Webster and Phillips, 2024), just as scaffolding is removed after a building is finished. Searching for a role for epimutation in adaptation may therefore involve studying the very early stages of the evolutionary process, perhaps using laboratory evolution experiments.

An interesting alternative scenario that could lead to long-term maintenance of an epimutation is if it compensated for a deleterious genetic change. Under such circumstances, the epimutation would have a highly beneficial selection coefficient. In theory, this could lead to long-term maintenance of the epimutation within the population. It is possible to imagine molecular scenarios that could give rise to this: for example, a mutation that decreased protein stability could be compensated by loss of repressive chromatin leading to higher transcription rates; alternatively toxic aggregation of a variant could be suppressed by increased epigenetic silencing. However, acquisition of mutations that had the same effect as the epimutation in suppressing the original genetic change would still be expected to go to fixation were they to arise, so even this scenario would still mean that the epimutation would eventually be replaced by genetic change.

## Conclusion and future directions

Whilst much is left to be determined experimentally, a series of tentative conclusions may be drawn about the role of epigenetic changes as drivers of the evolutionary processes.

1. Epimutations are unlikely to go to fixation in populations by drift due to their low heritability relative to genetic changes in the DNA sequence.
2. Strong selective benefit associated with an epimutation might promote its survival within populations.
3. Even if there is a strong selective benefit associated with an epimutation, it is likely that a genetic mutation at the same locus will eventually replace the epimutation.
4. The role of epimutation in evolution is therefore likely to be to accelerate adaptation through facilitation of genetic change. Potentially, this may involve direct stimulation of mutation due to certain types of epimutations.

These conclusions point the way forward for future experiments to close the gaps between the theoretical considerations presented here and empirical data. Epimutations may be very hard to detect in extant populations but instead will require careful analysis of evolving systems, which is best performed in the laboratory. A key prediction would be that disabling epigenetic pathways, particularly those associated with epimutations such as small noncoding RNA-mediated silencing in *C. elegans*, might slow evolutionary change. Equivalently, hyperactivation of epigenetic pathways, to give rise to an epimutator strain (Perales et al, 2018) might lead to faster adaptation to stressful environments. A further, important, prediction is that genetic mutations with similar effects to the epimutation would occur after the epimutation and rapidly replace it. Observing this in the laboratory is probably the single most important goal for the field in the near future.

Of course, the very long-term goal will be to apply these ideas to evolution in real populations in nature. Already, some plausible inferences have linked epimutations in *Arabidopsis* to divergence associated with different environments (Baduel et al, 2025); moreover, it seems likely that epimutations that promote resistance to multiple xenobiotics, including toxic drugs, exist at low frequencies in fission yeast populations (Fellas et al, 2024). However, there will inevitably be genetic mutations that co-occur with epigenetic differences in real populations, making the inference of causality complicated. Instead, better understanding of the regions where epimutations with phenotypic effects are most frequent in the laboratory should prompt investigation of these regions in natural populations to investigate whether the frequency of genetic mutations with similar predicted effects as epimutations is elevated.

Epigenetic systems evolve rapidly; even within animals, many highly conserved epigenetic mechanisms have been lost multiple times in different lineages (Sarkies et al, 2015; Sarkies, 2022; de Mendoza et al, 2020). It will be fascinating to uncover whether changes in epigenetic components, affecting the propensity for organisms to epimutate, have influenced the rate of evolution of different species, thus shaping processes such as speciation that underpin the richness of life on earth.

## Peer review information

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

## Acknowledgements

We thank Toby Warnecke (University of Oxford) for comments on the manuscript draft and Max Fitz-James and all other members of the Sarkies laboratory for helpful discussions. Work in the Sarkies laboratory is funded by the Leverhulme Trust and the Wellcome Trust.

## Author contributions

**Nabeel S Ganem**: Visualization; Writing—original draft; Writing—review and editing. **Peter Sarkies**: Conceptualization; Formal analysis; Visualization; Writing—original draft.

## Disclosure and competing interests statement

PS is a member of the *EMBO Press* Ecology and Evolution Advisory Board.

