## [Peer Review File · The EMBO Journal]

Epimutations: raw material for evolution?

Peter Sarkies and Nabeel Ganem

Corresponding author: Peter Sarkies (peter.sarkies@bioch.ox.ac.uk)

Review Timeline:

Submission Date:	29th Sep 25
Editorial Decision:	27th Oct 25
Revision Received:	4th Dec 25
Editorial Decision:	9th Dec 25
Revision Received:	14th Dec 25
Accepted:	17th Dec 25

Editor: Yehu Moran

Transaction Report:

Dear Peter,

Thank you once again for writing this insightful manuscript. The reviews are in, and while they are generally very positive, the three referees do raise quite a few comments. Hence, I would like to invite you to consider these comments and address them via a revision.

As we are short in time for meeting the deadline of the special issue, I would appreciate it very much if you can proceed with the review process within one month. I remain available for any consultations that might be helpful for you during this process.

Thank you very much for the submission and I look forward to your revision.

Yours sincerely,

Yehu Moran
Academic Editor
The EMBO Journal

We realize that it is difficult to revise to a specific deadline. In the interest of protecting the conceptual advance provided by the work, we recommend a revision within 3 months (25th Jan 2026). Please discuss the revision progress ahead of this time with the editor if you require more time to complete the revisions.

Referee #1:

In this review, Ganem and Sarkies examine how epimutations-heritable but reversible epigenetic changes such as DNA methylation, histone modifications, and small RNAs-might contribute to evolutionary processes. They define epimutations as molecular alterations that can change gene expression states without altering the DNA sequence and that can occasionally persist across generations.

The authors first outline mechanistic sources of epimutations, emphasizing examples from *C. elegans* small RNA inheritance, and then discuss their transmission dynamics and erasure. The second half of the review integrates these mechanisms into population-genetic frameworks, comparing the rates, stability, and phenotypic effects of epimutations with those of classical DNA mutations. The authors argue that, although epimutations are often transient, they may still influence adaptation by providing short-term heritable variation or by facilitating subsequent genetic change.

Overall, the review is clear, well-organized, and intellectually stimulating, offering a valuable conceptual bridge between molecular epigenetics and evolutionary theory. Some areas, however, could benefit from broader mechanistic coverage and additional examples to reinforce their general conclusions.

Specific Comments

1. Overemphasis on the *C. elegans* small-RNA paradigm

The mechanistic overview focuses almost exclusively on small RNA inheritance in *C. elegans*, giving the impression that TEI in animals is largely RNA-based. The authors should broaden the perspective to include chromatin-based inheritance mechanisms and clarify that not all epigenetic marks are erased during germline reprogramming.

In particular, the statement that "inheritance of altered histone modifications has not been shown to underpin transgenerational epigenetic inheritance in any animal" should be corrected. Several studies demonstrate inheritance and functional consequences of histone marks across generations. In *Drosophila*, Zenk et al. (Science 2017) and Atinbayeva et al. (EMBO J 2024) have shown that H3K9 and H3K27 methylation are inherited. Similarly, in *C. elegans*, Gaydos et al. (Science 2014) demonstrated inheritance of H3K27me3, and sperm have been shown to contribute histone modifications (Kaneshiro et al., Nat Commun 2019). Regarding phenotypic inheritance linked to histone methylation, work in *C. elegans* has shown that longevity and germline mortality phenotypes are associated with mutations in histone methyltransferases.

2. Definition and modeling of epimutations

The binary (on/off) model is convenient but biologically simplistic. Chromatin marks often exist in graded states, complicating estimates of rates and selection. The authors should note this limitation and distinguish stochastic from environmentally induced epimutations, as their evolutionary implications differ.

3. Epimutations as scaffolds for genetic change

The notion that epimutations "scaffold" subsequent genetic change is interesting but remains speculative. Examples of epigenetic states that precede genetic fixation (e.g., transposon mobilization, stress-induced mutagenesis) would strengthen the argument, as would discussion of ecological contexts in which reversible epimutations might confer adaptive flexibility.

4. Frequency versus effect size: limited impact of RdRP-dependent small-RNA epimutations

Although *C. elegans* RdRP-dependent small RNA "epimutations" occur frequently, transcriptomic data show that mutants lacking core components (*mut-16*, *prg-1*, *hrde-1*, *drh-3*, etc.) exhibit only modest and locus-restricted up-regulation. Hence, high frequency does not imply a strong phenotypic or evolutionary effect. The authors should explicitly separate epimutation rate from effect magnitude and note that most such events likely carry weak selection coefficients.

5. Treatment of plant examples

The authors state that their main focus is on animals, using plants and fungi "to illustrate key theoretical principles." However, plants provide the most compelling empirical demonstrations of stable, selectable epimutations, the very phenomenon under discussion. It would be valuable to include a few plant examples as empirical anchors, rather than merely theoretical illustrations, to provide balance and realism.

Referee #2:

Overall, I found this to be an insightful and thought-provoking review that applies population evolutionary genetics logic to unpick how epimutations might contribute to the adaptation, and thus the evolution of organisms.

In some parts it is perhaps a bit too theoretical but that is the perspective from which the authors are coming from and where they do a good job in relating theory to experimental observations.

Comments to consider (no need to act on these):

Other examples of epimutation in fungi such as *Mucor* that could be included:

Perex-Arques et al. *Nat Commun.* 2025 Aug 7;16(1):7293. doi: 10.1038/s41467-025-62572-6

Son and Heitman. *bioRxiv [Preprint]*. 2025 Jul 11:2025.06.17.660219. doi: 10.1101/2025.06.17.660219

Calo et al. *Nature*. 2014 Sep 25;513(7519):555-8. doi: 10.1038/nature13575. Epub 2014 Jul 27.

The review focusses on transgenerational inheritance of epimutations but it may be worth mentioning that epimutations can also influence phenotypes in somatic cells where erasure/resetting does not occur - especially cancer cells. See for example: *Cell*. 2020 Aug 20;182(4):947-959.e17. doi: 10.1016/j.cell.2020.07.003.

Page 4-5:

It is incorrect to refer to protamines as specialised nucleosomes. Structurally they are a completely different from the four histones that make a nucleosome.

Many studies indicate that histones and histone PTMs can be retained in sperm, the variation may or may not be linked to subsequent phenotypic/developmental variation. See review that has good discussion: *Curr Opin Genet Dev.* 2023 Apr;79:102034. doi: 10.1016/j.gde.2023.102034.

Apart from *C. elegans* do other animals show RNAi-mediated transgenerational inheritance? If so, it would be worth mentioning these other examples. Related, what other animals retain active RdRP? Does the presence/absence of RdRP predict mechanistic confinement to those animals that retain it?

The authors might want to consider and include the recent finding that imposing DNA methylation triggers the transgenerational inheritance of that DNA methylation and associated phenotype in mice.

Cell. 2023 Feb 16;186(4):715-731.e19. doi: 10.1016/j.cell.2022.12.047. Epub 2023 Feb 7.

Page 5/para 2:

The term amplification perhaps implies that multiple copies of the signalling entity that is transmitted between generations needs to be made to allow its inheritance. This is clearly the case with RNAi however, a robust read-write mechanism that just accurately copies the pre-existing DNA/histone modification would also suffice.

Page6/para2:

RNAi pathways do direct DNA methylation in *Arabidopsis*.

If chromatin modifications are directly linked to small RNAs in *C. elegans* then sRNAs could decline below detectability but reappear due to the chromatin state. In this case both would need to be quantified, not just sRNAs - presumably resulting models that define such epimutations would be more complex?

The authors should make it more obvious that RNAi and DNA/chromatin modifications are often co-dependent or dependent processes.

Page7/para1:

The authors mention that epimutations defined by regions exhibiting less ATAC-accessibility and sRNAs in *C. elegans* are more stable. Is it known how accessibility and sRNA are connected at these locations? Do genes in these regions have a higher overall mutation rate? Is it possible that these processes promote genetic mutation to increase the probability of a beneficial mutation emerging?

/para4

Why do piRNAs provide a better indicator of epimutations?

Page 8/para 1

Xenobiotic defence: see Fellas et al *BioRxiv* doi.org/10.1101/2024.11.13.623381 wild-type fission yeast cells harness epimutations that impose mitochondrial dysfunction to bypass external insults. Here the epimutation reduces expression of a gene encoding an essential mitochondrial function to allow cells to resist exposure to a toxin.

Page 8-10 The effects of epimutations

It may be worth noting that an epimutation can act as a rheostat so that the level of expression of a gene/genes can vary from cell to cell or individual to individual allowing some cells/individuals to exhibit optimal expression levels for survival in condition X while others will be at a selective disadvantage. A greater fitness cost may be associated with a stronger epimutation (e.g. lower expression) at the same locus thus such strong epimutations are selected against while weaker optimal epimutations are advantageous in condition X.

In addition, what is deleterious in condition X may be advantageous in condition Y (see Fellas et al above).

Page 10/para2

'biased genome-wide distribution of epimutations': it is highly likely that the millions of years of evolution during which an organism has been subjected to many different environmental challenges has shaped where advantageous epimutations are most likely to form. Repeated rounds of selection over evolutionary time must increase the epigenetic plasticity of such regions

and presumably specific features of DNA in, or RNA made from, these regions must influence their propensity to form epimutations.

Page 11:

'harder for an epimutation to influence natural selection than a genetic epimutation':

the distinct advantage of an epimutation, regardless of mechanism or duration, is that it allows a subpopulation of wild type individuals to make it through a selective bottleneck without imposing a permanent fitness cost that may be associated with a genetic change (no matter how marginal that cost might be). Thus, epimutations introduce heterogeneity and allow bet-hedging within a genetically identical population, that is subject to a changeable environment, without compromising longer-term genetic integrity. An advantageous mutation could result in a deleterious dead-end whereas an unstable epimutation would allow the reemergence of wild-type individuals in the absence of selective conditions.

Page 12 top:

'shorter because there are more individuals': I understand the case being made here but the phrase 'accelerate the process of genetic change within the population' is perhaps a bit confusing.

Obviously under selective conditions a beneficial epimutation will allow more individuals to survive, therefore because there are more individuals this population increases and continued selection will select for an advantageous genetic mutation. 'accelerate the process of genetic change within the population' might be confused with the epimutation itself influencing the mutation rate at the epimutated locus itself.

Stress that the frequency of mutation itself is not altered.

Corrections:

Intro/para2: (genetic diff) (Blount et al 2018). >drift

Page 12: hba-1 should be hba1 in italics

Referee #3:

The manuscript provides a broad overview of epigenetic inheritance and related concepts. While the topic is timely and potentially valuable, the manuscript can be made more effective by reducing redundancy and improving focus. For example, the sentence 'An important point to make right at the outset...' is unnecessarily wordy and could be expressed much more directly.

At present, the text is quite long and reads somewhat like an extended introduction chapter. It attempts to provide historical context, but this is uneven. The first sentence: "...the past decade has revealed...." shows that some of the early work from >25 years ago is omitted that introduces some of the concepts delineated here (examples DOI: 10.1006/jtbi.1999.0974 , <https://doi.org/10.1016/j.gde.2004.09.001>, [https://doi.org/10.1016/s0960-9822\(02\)01377-5](https://doi.org/10.1016/s0960-9822(02)01377-5) , reviewed in <https://www.nature.com/articles/s41576-021-00438-5> . Other sections go into more detail than needed. A more selective and focused narrative would improve readability and impact.

I would encourage the authors to concentrate on one central aspect of the topic that they want to promote. For example an area where they can provide particular insight, such as the estimation of epigenetic mutation accumulation (eMA). Focusing on a specific question would help the review develop a clearer through-line and a stronger argument.

For those outside the immediate area of epigenetic inheritance, some parts of the manuscript may be challenging to follow. To aid readability, it might help to include a concise text box summarising definitions, assumptions, and key terms. For example, just define how the term 'transgenerational inheritance' is used (greater than or equal to 3 generations). This would make the review more accessible to readers less familiar with the field, while allowing the main text to stay streamlined.

Overall, the paper has the potential to be an informative and useful review once the scope is narrowed and the argument sharpened.

Some specific comments:

- Figure one talks about 'Argonaute proteins' but they are not depicted in the illustration and in the body of the text they are simply described as 'stabilizing factors of 22G-RNAs.
- "This is because of an enzyme, maintenance methyltransferase (DNMT1 in animals) that recognizes hemimethylated DNA that forms during replication of DNA containing 5mC(Song et al., 2011). It then introduces methylation onto the other strand, thus perpetuating the original modification state.."

This point could be clarified for readers less familiar with the concept - namely, that the parental DNA strand carries the epigenetic mark (e.g., 5mC), which is subsequently copied to the newly synthesised daughter strand by DNMT1 during replication.

- " However, in mammals, this is unlikely because most methylation is wiped out at two different stages..." There may be more

precise or nuanced ways to describe the dynamics of DNA methylation in mammals.

- Similarly, the authors might consider refining this phrasing to more accurately "....It is important to note that the wholesale eradication of DNA methylation in mammals is not widely conserved ..."
- ".... However, RdRPs do exist in many other animals(Lewis et al., 2018), thus could contribute to transgenerational epigenetic inheritance in other organisms besides nematodes...." Do the authors mean in "other animal clades...?"

Referee #1:

In this review, Ganem and Sarkies examine how epimutations-heritable but reversible epigenetic changes such as DNA methylation, histone modifications, and small RNAs-might contribute to evolutionary processes. They define epimutations as molecular alterations that can change gene expression states without altering the DNA sequence and that can occasionally persist across generations.

The authors first outline mechanistic sources of epimutations, emphasizing examples from *C. elegans* small RNA inheritance, and then discuss their transmission dynamics and erasure. The second half of the review integrates these mechanisms into population-genetic frameworks, comparing the rates, stability, and phenotypic effects of epimutations with those of classical DNA mutations. The authors argue that, although epimutations are often transient, they may still influence adaptation by providing short-term heritable variation or by facilitating subsequent genetic change.

Overall, the review is clear, well-organized, and intellectually stimulating, offering a valuable conceptual bridge between molecular epigenetics and evolutionary theory. Some areas, however, could benefit from broader mechanistic coverage and additional examples to reinforce their general conclusions.

Specific Comments

1. Overemphasis on the *C. elegans* small-RNA paradigm

The mechanistic overview focuses almost exclusively on small RNA inheritance in *C. elegans*, giving the impression that TEI in animals is largely RNA-based. The authors should broaden the perspective to include chromatin-based inheritance mechanisms and clarify that not all epigenetic marks are erased during germline reprogramming.

In particular, the statement that "inheritance of altered histone modifications has not been shown to underpin transgenerational epigenetic inheritance in any animal" should be corrected. Several studies demonstrate inheritance and functional consequences of histone marks across generations. In *Drosophila*, Zenk et al. (Science 2017) and Atinbayeva et al. (EMBO J 2024) have shown that H3K9 and H3K27 methylation are inherited. Similarly, in *C. elegans*, Gaydos et al. (Science 2014) demonstrated inheritance of H3K27me₃, and sperm have been shown to contribute histone modifications (Kaneshiro et al., Nat Commun 2019). Regarding phenotypic inheritance linked to histone methylation, work in *C. elegans* has shown that longevity and germline mortality phenotypes are associated with mutations in histone methyltransferases.

>>We agree that it is important to note that transgenerational epigenetic inheritance of histone modifications occurs, this was mentioned but dealt with briefly due to space limitations. In the revised version we have covered this in more depth, including examples from model organisms where histone post-translational modification state changes have been shown to be transmitted transgenerationally. Formal analysis of **epimutations** genome-wide due to these processes have not been published yet, however.

2. Definition and modeling of epimutations

The binary (on/off) model is convenient but biologically simplistic. Chromatin marks often exist in graded states, complicating estimates of rates and selection. The authors should note this limitation and distinguish stochastic from environmentally induced epimutations, as their evolutionary implications differ.

>>We have now included a section on why positive feedback likely makes most epimutations bimodal, including reference to histone modifications, where read-write mechanisms will tend

towards bimodality as well as small non-coding RNAs and DNA methylation. The bimodality is a property shared by all types of epimutations, and this has been the focus of the modelling and analysis that has been done so far, so our current article must emphasize this. However, we do acknowledge in the text that there may be more complex cases where a quantitative response occurs.

We don't really agree, however, that there is necessarily a fundamental distinction in whether environmentally induced or stochastic epimutations are more likely to be bimodal and decided not to enter into a discussion of this in the manuscript for space reasons.

3. Epimutations as scaffolds for genetic change

The notion that epimutations "scaffold" subsequent genetic change is interesting but remains speculative. Examples of epigenetic states that precede genetic fixation (e.g., transposon mobilization, stress-induced mutagenesis) would strengthen the argument, as would discussion of ecological contexts in which reversible epimutations might confer adaptive flexibility.

>>We agree that it is speculative. However, the mechanistic basis for this is very clear, as we discuss, because repressive chromatin is well known to impact mutation rate. We agree that transposon mobilisation is a good possible example to include, and so have introduced this as a new paragraph drawing on data from plants on the effects of epimutations on transposon mobilisation.

4. Frequency versus effect size: limited impact of RdRP-dependent small-RNA epimutations

Although *C. elegans* RdRP-dependent small RNA "epimutations" occur frequently, transcriptomic data show that mutants lacking core components (*mut-16*, *prg-1*, *hrde-1*, *drh-3*, etc.) exhibit only modest and locus-restricted up-regulation. Hence, high frequency does not imply a strong phenotypic or evolutionary effect. The authors should explicitly separate epimutation rate from effect magnitude and note that most such events likely carry weak selection coefficients.

>>We completely agree with the reviewer and in fact there was already a discussion of this point in the original version of the manuscript, explaining how the distribution of fitness effects is likely to mean that epimutations have a very small effect on fitness on average.

5. Treatment of plant examples

The authors state that their main focus is on animals, using plants and fungi "to illustrate key theoretical principles." However, plants provide the most compelling empirical demonstrations of stable, selectable epimutations, the very phenomenon under discussion. It would be valuable to include a few plant examples as empirical anchors, rather than merely theoretical illustrations, to provide balance and realism.

>>We have now included some more plant examples to illustrate certain points, particularly transposon mobilisation due to epimutations (see above) and the potential of some epimutations to remain in the population due to selection pressure as this has recently been described in plants.

Referee #2:

Overall, I found this to be an insightful and thought-provoking review that applies population evolutionary genetics logic to unpick how epimutations might contribute to the adaptation, and thus the evolution of organisms.

In some parts it is perhaps a bit too theoretical but that is the perspective from which the authors are coming from and where they do a good job in relating theory to experimental observations.

Comments to consider (no need to act on these):

Page 2/Intro Para3

Other examples of epimutation in fungi such as *Mucor* that could be included:

Perex-Arques et al. *Nat Commun.* 2025 Aug 7;16(1):7293. doi: 10.1038/s41467-025-62572-6

Son and Heitman. *bioRxiv [Preprint]*. 2025 Jul 11:2025.06.17.660219. doi:

10.1101/2025.06.17.660219

Calo et al. *Nature.* 2014 Sep 25;513(7519):555-8. doi: 10.1038/nature13575. Epub 2014 Jul 27.

>>Thank you, these have been included in the revised version.

The review focusses on transgenerational inheritance of epimutations but it may be worth mentioning that epimutations can also influence phenotypes in somatic cells where erasure/resetting does not occur - especially cancer cells. See for example: *Cell.* 2020 Aug 20;182(4):947-959.e17. doi: 10.1016/j.cell.2020.07.003.

>>This is certainly true, but outside the scope of the review. We have clarified this in the introduction and included a link to a recent review on the topic (<https://www.mdpi.com/2072-6694/13/6/1380>) rather than picking one example.

Page 4-5:

It is incorrect to refer to protamines as specialised nucleosomes. Structurally they are a completely different from the four histones that make a nucleosome.

>>thank you, we have corrected this

Many studies indicate that histones and histone PTMs can be retained in sperm, the variation may or may not be linked to subsequent phenotypic/developmental variation. See review that has good discussion: *Curr Opin Genet Dev.* 2023 Apr;79:102034. doi:

10.1016/j.gde.2023.102034.

>>Thank you, this was noted in the original version but we have altered the text slightly to emphasize this point and cite the suggested review.

Apart from *C. elegans* do other animals show RNAi-mediated transgenerational inheritance? If so, it would be worth mentioning these other examples. Related, what other animals retain active RdRP? Does the presence/absence of RdRP predict mechanistic confinement to those animals that retain it?

>>We have no idea. Indeed, it's not even clear whether transgenerational epigenetic inheritance is even common across nematodes, let alone in species where we don't know what RdRP is doing. We have noted this point in the manuscript.

The authors might want to consider and include the recent finding that imposing DNA methylation triggers the transgenerational inheritance of that DNA methylation and associated phenotype in mice.

Cell. 2023 Feb 16;186(4):715-731.e19. doi: 10.1016/j.cell.2022.12.047. Epub 2023 Feb 7.

>>Thank you, we agree that this is a nice illustration of the conceptual possibility of transgenerational epigenetic inheritance due to DNA methylation changes and we have now included this in the revised version.

Page 5/para 2:

The term amplification perhaps implies that multiple copies of the signalling entity that is transmitted between generations needs to be made to allow its inheritance. This is clearly the case with RNAi however, a robust read-write mechanism that just accurately copies the pre-existing DNA/histone modification would also suffice.

>>We would argue that histone post-translational modifications and DNA methylation still could be described as amplification, because the level that remains after replication is amplified by the read/write mechanisms, thus approximately duplicating the post-replicative level to restore levels of histone modifications present before DNA replication. We have now explained mechanistically the basis of histone post-translational modification maintenance, which we hope clarifies this.

Page6/para2:

RNAi pathways do direct DNA methylation in Arabidopsis.

If chromatin modifications are directly linked to small RNAs in *C. elegans* then sRNAs could decline below detectability but reappear due to the chromatin state. In this case both would need to be quantified, not just sRNAs - presumably resulting models that define such epimutations would be more complex?

True but in *C. elegans* it is almost certainly the sRNAs that are transmitted not the chromatin modifications. We have clarified this in the manuscript as well as citing the relevant evidence that confirms direct transfer of sRNAs between generations in *C. elegans*

The authors should make it more obvious that RNAi and DNA/chromatin modifications are often co-dependent or dependent processes.

Thank you for this important point, which we have now emphasized in the relevant section on mechanism of transgenerational epigenetic inheritance.

Page7/para1:

The authors mention that epimutations defined by regions exhibiting less ATAC-accessibility and sRNAs in *C. elegans* are more stable. Is it known how accessibility and sRNA are connected at these locations? Do genes in these regions have a higher overall mutation rate? Is it possible that these processes promote genetic mutation to increase the probability of a beneficial mutation emerging?

>>These are all interesting questions but the answers are not known. We discuss the idea of genetic mutation promoted by epimutations, but this is currently speculation on our part.

/para4

Why do piRNAs provide a better indicator of epimutations?

>>22GRNAs are a good indicator of epimutations because they tend to behave bimodally at genes- either high or low levels. piRNAs can't easily be mapped to individual genes because they target with many mismatches, while 22GRNAs have exactly complementary sequence to their targets due to their biogenesis mechanism dependent on RdRP.

Page 8/para 1

Xenobiotic defence: see Fellas et al BioRxiv doi.org/10.1101/2024.11.13.623381 wild-type fission yeast cells harness epimutations that impose mitochondrial dysfunction to bypass external insults. Here the epimutation reduces expression of a gene encoding an essential mitochondrial function to allow cells to resist exposure to a toxin.

>>Thank you, we have included this in the conclusion section, where it seemed to fit best since prior to this, the fission yeast example had not yet been considered in detail.

Page 8-10 The effects of epimutations

It may be worth noting that an epimutation can act as a rheostat so that the level of expression

of a gene/genes can vary from cell to cell or individual to individual allowing some cells/individuals to exhibit optimal expression levels for survival in condition X while others will be at a selective disadvantage. A greater fitness cost may be associated with a stronger epimutation (e.g. lower expression) at the same locus thus such strong epimutations are selected against while weaker optimal epimutations are advantageous in condition X. In addition, what is deleterious in condition X may be advantageous in condition Y (see Fellas et al above).

>>Thank you; this is of course plausible, but we tend to think that epimutations are mostly going to be bimodal due to their mechanistic properties (see discussion of this point in the response to reviewer 1).

Page 10/para2

'biased genome-wide distribution of epimutations': it is highly likely that the millions of years of evolution during which an organism has been subjected to many different environmental challenges has shaped where advantageous epimutations are most likely to form. Repeated rounds of selection over evolutionary time must increase the epigenetic plasticity of such regions and presumably specific features of DNA in, or RNA made from, these regions must influence their propensity to form epimutations.

>>It is not obvious that this is the case because much depends on how strong selection is for/against epimutations. If epimutations have very little fitness impact selection is unlikely to alter their genome-wide distribution. It could simply be to do with the mechanisms whereby epimutations occur that make some genomic regions more susceptible. We have clarified this point in the text.

Page 11:

'harder for an epimutation to influence natural selection than a genetic epimutation': the distinct advantage of an epimutation, regardless of mechanism or duration, is that it allows a subpopulation of wild type individuals to make it through a selective bottleneck without imposing a permanent fitness cost that may be associated with a genetic change (no matter how marginal that cost might be). Thus, epimutations introduce heterogeneity and allow bet-hedging within a genetically identical population, that is subject to a changeable environment, without compromising longer-term genetic integrity. An advantageous mutation could result in a deleterious dead-end whereas an unstable epimutation would allow the reemergence of wild-type individuals in the absence of selective conditions.

>>Thank you. We agree with this mechanism in theory, but nevertheless, the ability of this to actually work in practise depends on the inheritance and fitness consequences of epimutations, as we discuss in Box 1.

Page 12 top:

'shorter because there are more individuals': I understand the case being made here but the phrase 'accelerate the process of genetic change within the population' is perhaps a bit confusing.

Obviously under selective conditions a beneficial epimutation will allow more individuals to survive, therefore because there are more individuals this population increases and continued selection will select for an advantageous genetic mutation. 'accelerate the process of genetic change within the population' might be confused with the epimutation itself influencing the mutation rate at the epimutated locus itself.

Stress that the frequency of mutation itself is not altered.

>>Thank you. We were trying to make this point in the original version, but were not clear enough. We've tightened the wording in the revised version.

Corrections:

Intro/para2: (genetic drift) (Blount et al 2018). >drift

Page 12: hba-1 should be hba1 in italics

>>thank you, we have corrected these points.

Referee #3:

The manuscript provides a broad overview of epigenetic inheritance and related concepts. While the topic is timely and potentially valuable, the manuscript can be made more effective by reducing redundancy and improving focus. For example, the sentence 'An important point to make right at the outset...' is unnecessarily wordy and could be expressed much more directly.

At present, the text is quite long and reads somewhat like an extended introduction chapter. It attempts to provide historical context, but this is uneven. The first sentence: "...the past decade has revealed...." shows that some of the early work from >25 years ago is omitted that introduces some of the concepts delineated here (examples DOI: 10.1006/jtbi.1999.0974, <https://doi.org/10.1016/j.gde.2004.09.001>, [https://doi.org/10.1016/S0960-9822\(02\)01377-5](https://doi.org/10.1016/S0960-9822(02)01377-5), reviewed in <https://www.nature.com/articles/S41576-021-00438-5>. Other sections go into more detail than needed. A more selective and focused narrative would improve readability and impact.

I would encourage the authors to concentrate on one central aspect of the topic that they want to promote. For example an area where they can provide particular insight, such as the estimation of epigenetic mutation accumulation (eMA). Focusing on a specific question would help the review develop a clearer through-line and a stronger argument.

>>We thank the reviewer for their comments. However, we think that we have already written a focussed article by concentrating on epimutations rather than considering the many other ways in which epigenetics and genetics interact in evolution. We do agree that some of the wording was overly verbose, so have attempted to remove some of the redundancy in the phrasing to improve readability.

For those outside the immediate area of epigenetic inheritance, some parts of the manuscript may be challenging to follow. To aid readability, it might help to include a concise text box summarising definitions, assumptions, and key terms. For example, just define how the term 'transgenerational inheritance' is used (>{greater than or equal to} 3 generations).

>>We have now more clearly defined transgenerational inheritance to avoid confusion.

This would make the review more accessible to readers less familiar with the field, while allowing the main text to stay streamlined.

>>We decided not to include a glossary because we think it's better to define the words in the text to avoid disrupting flow.

Overall, the paper has the potential to be an informative and useful review once the scope is narrowed and the argument sharpened.

>>Thank you, we hope that with the revisions we have sharpened some of the arguments.

Some specific comments:

- Figure one talks about 'Argonaute proteins' but they are not depicted in the illustration and in the body of the text they are simply described as 'stabilizing factors of 22G-RNAs.

>>Thank you, we have changed this

- "This is because of an enzyme, maintenance methyltransferase (DNMT1 in animals) that recognizes hemimethylated DNA that forms during replication of DNA containing 5mC(Song et

al., 2011). It then introduces methylation onto the other strand, thus perpetuating the original modification state.."

This point could be clarified for readers less familiar with the concept - namely, that the parental DNA strand carries the epigenetic mark (e.g., 5mC), which is subsequently copied to the newly synthesised daughter strand by DNMT1 during replication.

>>Thank you, this is a helpful suggestion.

- " However, in mammals, this is unlikely because most methylation is wiped out at two different stages..." There may be more precise or nuanced ways to describe the dynamics of DNA methylation in mammals.

>>Thank you, we have made this more precise.

- Similarly, the authors might consider refining this phrasing to more accurately "...It is important to note that the wholesale eradication of DNA methylation in mammals is not widely conserved ..."

>>We are not sure what reviewer means by this

- "... However, RdRPs do exist in many other animals(Lewis et al., 2018), thus could contribute to transgenerational epigenetic inheritance in other organisms besides nematodes..." Do the authors mean in "other animal clades...?"

>>Thank you, this is what we meant and have clarified in the manuscript.

Dear Prof. Sarkies,

Thank you for submitting your revised manuscript for consideration by the EMBO Journal. I have now had the chance to review your revision and I provide my feedback below. I am also providing the feedback of our editorial assistance team that is more technical in its nature.

When preparing your letter of response to my comments, please bear in mind that this will form part of the Review Process File, and will therefore be available online to the community. For more details on our Transparent Editorial Process, please review our Editorial Policies page: <https://link.springer.com/partners/embo-press/editorial-policies>
There is no need to address the editorial assistance team comments in a point-by-point format, but it is essential to accommodate their requested edits as this is essential for official acceptance.

Please try to return your revised manuscript by the end of the year.

Thank you for the opportunity to consider your work for publication. I look forward to your final revision.

Yours sincerely,

Yehu Moran
Academic Editor
The EMBO Journal

Read our guidance for manuscript revisions and related editorial policies: <https://link.springer.com/journal/44318/submission-guidelines#cms-Revised-submissions>

<https://media.springernature.com/original/springer-cms/rest/v1/content/27825798/data/v1>

- a point-by-point response to the referees' comments, with a detailed description of the changes made (as a word file).
- a word file of the manuscript text.
- individual production quality figure files (one file per figure)
- a complete author checklist
- Expanded View files (replacing Supplementary Information)
- a Reagents and Tools Table as part of the Methods section

Please remember: Digital image enhancement is acceptable practice, as long as it accurately represents the original data and conforms to community standards. If a figure has been subjected to significant electronic manipulation, this must be noted in the figure legend or in the 'Methods' section. The editors reserve the right to request original versions of figures and the original images that were used to assemble the figure.

We realize that it is difficult to revise to a specific deadline. In the interest of protecting the conceptual advance provided by the work, we recommend a revision within 3 months (9th Mar 2026). Please discuss the revision progress ahead of this time with the editor if you require more time to complete the revisions.

comments by academic editor (Yehu Moran)

When the authors mention that RdRps are present in other animals, it would be good to refer not only to Lewis et al. 2018 Nat. Ecol. Evol., but also to Zong et al. 2009 Gene doi: 10.1016/j.gene.2009.07.004

Typos to fix:

- * "a phenomenon known a transgenerational..." should be "known as transgenerational".
- * "in this review is to whether these kinds of effects..." should be "in this review is to assess whether these kinds of effects...".
- * "H327me3 domains" should be "H3K27me3 domains".
- * "much more likely for an to have beneficial effects" something seems to be missing here.
- * "epimutation and adaption by natural selection" I guess it should be "adaptation".

comments by editorial assistance team

- Keywords: missing, please provide up to five keywords.
- ACKNOWLEDGEMENTS/FUNDING: no funding mentioned in the manuscript but two funders are listed in the system: Leverhulme Trust and Wellcome Trust - please add these in an Acknowledgments section in the text.
- DISCLOSURE AND COMPETING INTERESTS STATEMENT: missing, please add, including a sentence that P.S. is a member of the Eco-Evo editorial advisory board of EMBO Press.
- FIGURES: please remove them from the manuscript text and merge the two parts of Figure 2 into a single file- we only allow one page per figure.
- FIGURE LEGENDS: please move to the end of the manuscript text.

The authors addressed the remaining editorial issues.

Dear Dr. Sarkies,

I am pleased to inform you that your manuscript has been accepted for publication in the EMBO Journal.

Your manuscript will be processed for publication by EMBO Press. It will be copy edited and you will receive page proofs prior to publication. Please note that you will be contacted by Springer Nature Author Services to complete licensing and payment information. As this review article was invited by us, we will provide you in a separate email with a waiver token you can put into the Springer-Nature system to fully cover the publication costs.

Yours sincerely,

Yehu Moran
Academic Editor
The EMBO Journal

Please note that it is The EMBO Journal policy for the transcript of the editorial process (containing referee reports and your response letters) to be published as an online supplement to each paper. If you should prefer removal of any referee-only figures included in the point-by-point response(s), e.g. because they may still be used for future publication or because they have been reproduced from published work by others, please do let us know immediately via response email.

More information is available here: <https://link.springer.com/partners/embo-press/editorial-policies#Peer%20review>